# Disorder of Sex Development Due to 17-Beta-Hydroxysteroid Dehydrogenase Type 3 Deficiency: A Case Report and Review of 70 Different *HSD17B3* Mutations Reported in 239 Patients

**DOI:** 10.3390/ijms231710026

**Published:** 2022-09-02

**Authors:** Catarina I. Gonçalves, Josianne Carriço, Margarida Bastos, Manuel C. Lemos

**Affiliations:** 1CICS-UBI, Health Sciences Research Centre, University of Beira Interior, 6200-506 Covilha, Portugal; 2Serviço de Endocrinologia, Diabetes e Metabolismo, Centro Hospitalar Universitário de Coimbra, 3000-075 Coimbra, Portugal; 3C4-UBI, Cloud Computing Competence Centre, University of Beira Interior, 6200-501 Covilha, Portugal

**Keywords:** 17-beta-hydroxysteroid dehydrogenase type 3, *HSD17B3*, Disorder of Sex Development (DSD), pseudohermaphroditism, mutation

## Abstract

The 17-beta-hydroxysteroid dehydrogenase type 3 (17-β-HSD3) enzyme converts androstenedione to testosterone and is encoded by the *HSD17B3* gene. Homozygous or compound heterozygous *HSD17B3* mutations block the synthesis of testosterone in the fetal testis, resulting in a Disorder of Sex Development (DSD). We describe a child raised as a female in whom the discovery of testes in the inguinal canals led to a genetic study by whole exome sequencing (WES) and to the identification of a compound heterozygous mutation of the *HSD17B3* gene (c.608C>T, p.Ala203Val, and c.645A>T, p.Glu215Asp). Furthermore, we review all *HSD17B3* mutations published so far in cases of 17-β-HSD3 deficiency. A total of 70 different *HSD17B3* mutations have so far been reported in 239 patients from 187 families. A total of 118 families had homozygous mutations, 63 had compound heterozygous mutations and six had undetermined genotypes. Mutations occurred in all 11 exons and were missense (55%), splice-site (29%), small deletions and insertions (7%), nonsense (5%), and multiple exon deletions and duplications (2%). Several mutations were recurrent and missense mutations at codon 80 and the splice-site mutation c.277+4A>T each represented 17% of all mutated alleles. These findings may be useful to those involved in the clinical management and genetic diagnosis of this disorder.

## 1. Introduction

17-beta-hydroxysteroid dehydrogenase type 3 (17-β-HSD3) deficiency (OMIM: 264300) is a rare autosomal recessive Disorder of Sex Development (DSD) [1]. 17-β-HSD3 is encoded by the *HSD17B3* gene (chromosome 9q22.32) and expressed in the testis where it converts the inactive steroid androstenedione to the active androgen testosterone [2]. Homozygous or compound heterozygous *HSD17B3* mutations block the synthesis of testosterone in the fetal testis, resulting in undervirilized genitalia in 46,XY newborns [3]. The clinical phenotype may range from the normal female appearance of the external genitalia to various degrees of genital ambiguity, including microphallus with hypospadias [1,4,5]. Due to normal anti-Mullerian hormone (AMH) secretion, patients lack Mullerian structures (uterus, fallopian tubes, cervix, and the upper part of the vagina), whereas Wolffian derivatives (epididymis, vas deferens, and seminal vesicles) are often normally developed and testes are located in the inguinal canal. As female external genitalia is common at birth, these children are usually assigned the female gender and raised as such [1,4,5]. The diagnosis may occur in childhood due to mild clitoromegaly, urogenital sinus or inguinal hernia with testes present along the inguinal canals or labioscrotal folds [1,4,5]. At the expected time of puberty, severe virilization usually occurs due to the significant increase in the testicular secretion of androstenedione that can presumably be partially converted to testosterone by 17-β-HSD3 residual activity or by other isoenzymes [6,7]. Thus, diagnosis before puberty allows early treatment by the removal of the abnormal testes, which should prevent the clinical signs of marked virilization.

The phenotype of 17-β-HSD3 deficiency is often clinically indistinguishable from other DSDs, namely that of androgen insensitivity syndrome and 5-α-reductase type 2 deficiency [8]. Measurements of basal and human chorionic gonadotropin (hCG)-stimulated levels of sex steroids, their precursors, and metabolites can help distinguish between these defects [8]. However, these biochemical studies are not always conclusive and molecular genetic studies can be used for a definitive diagnosis [9,10,11].

We present a case of DSD in whom a definitive diagnosis of 17-β-HSD3 deficiency was established only after whole exome sequencing (WES) revealed a compound heterozygous mutation of the *HSD17B3* gene. Furthermore, we present an update on all *HSD17B3* mutations reported so far in the literature.

## 2. Results

### 2.1. Clinical Studies

The patient was the first-born daughter of non-consanguineous Portuguese healthy parents. The pregnancy was uneventful and she was born full-term by Cesarean section with a weight, length, and head circumference of 2800 g, 49 cm, and 34 cm, respectively. She had normal development until the age of 5 years, when she was brought to medical attention due to an enlargement of the clitoris (approximately 2 cm length) and the presence of small bilateral inguinal masses. No other signs of virilization or pubertal development were present. A pelvic ultrasound exam and laparoscopy revealed the absence of uterus and ovaries. A biopsy of the inguinal masses revealed the presence of testicular tissue. A chromosomal analysis of peripheral blood revealed a male 46,XY karyotype. A 3-day human chorionic gonadotropin (hCG) stimulation test was performed, and this showed an increase in testosterone and dihydrotestosterone (DHT) plasma levels (Table 1). The testosterone to DHT ratio was normal. Androstenedione levels were not measured. A provisional clinical diagnosis of male pseudohermaphroditism due to partial androgen insensitivity was made. At this time, a genetic test through sequencing of the *AR* (Androgen Receptor) and the *SRD5A2* (Steroid 5-Alpha-Reductase 2) genes revealed no coding sequence mutations. The child and her parents received psychological counselling and it was decided that she would maintain a female gender identity. At the age of 6 years, the child underwent bilateral orchiectomy and corrective surgery for the clitoromegaly. At the age of 12 years, she initiated transdermal estrogen therapy for the induction of puberty. She maintained normal growth and development and was last observed at the age of 18 years when she was referred for vaginoplasty due to short (~3 cm) vaginal length. There was no known family history of DSD.

### 2.2. Genetic Studies

Due to the negative findings upon sequencing the *AR* and *SRD5A2* genes, the genetic studies were completed by WES, followed by the analysis of a virtual panel of genes that were previously associated with DSDs [11]. This revealed that the patient was a compound heterozygote for two missense variants in exon 9 of the *HSD17B3* gene (Figure 1). These variants consisted of NM_000197.2:c.608C>T, which changed an alanine to a valine at amino acid position 203 (p.Ala203Val), and NM_000197.2:c.645A>T, which changed a glutamic acid to an aspartic acid at amino acid position 215 (p.Glu215Asp). Both variants fulfilled the American College of Medical Genetics and Genomics (ACMG) criteria for “Pathogenic” [12] and have been reported as causative mutations in patients with 17-β-HSD3 deficiency [3,13].

### 2.3. Mutations Reported in the Literature

Sixty-eight articles reported one or more patients with *HSD17B3* mutations (Appendix A) [1,3,4,5,9,10,13,14,15,16,17,18,19,20,21,22,23,24,25,26,27,28,29,30,31,32,33,34,35,36,37,38,39,40,41,42,43,44,45,46,47,48,49,50,51,52,53,54,55,56,57,58,59,60,61,62,63,64,65,66,67,68,69,70,71,72,73,74]. A total of 187 families (239 patients) were reported, of which 118 had homozygous mutations, 63 had compound heterozygous mutations and six had heterozygous mutations without data on the second mutation. In the 187 families (374 alleles), the types of mutations were missense (55%), splice-site (29%), small deletions and insertions (7%), nonsense (5%), and multiple exon deletions and duplications (2%). Mutations were distributed across all 11 exons (Figure 2). Many mutations were recurrent (i.e., shared by two or more families) and some occurred at unusually high frequencies: missense mutations at codon 80 and the splice-site mutation c.277+4A>T, each represented 17% of all mutated alleles. Altogether, there have been a total of 70 different *HSD17B3* mutations reported so far in the literature (Appendix A).

## 3. Discussion

We describe the clinical course and laboratory findings of a child with DSD in whom a genetic analysis revealed 17-β-HSD3 deficiency. The differential diagnosis with other similar 46,XY DSDs requires a high level of clinical suspicion and laboratory evaluation of the different hormones, their precursors, and metabolites that characterize each disorder, with confirmation by a molecular mutation analysis [8].

Patients with 17-β-HSD3 deficiency can vary in the appearance of their external genitals, which may be due to different residual enzyme activities [1,4,5]. In most cases, the 46,XY neonate presents apparently normal external female genitalia and is raised as a female [1,4,5]. In patients with lumps in the inguinal canals or labioscrotal folds, the palpation of gonads may lead to an early diagnosis, similar to our patient. If undiagnosed, these patients usually present during puberty with primary amenorrhea and varying degrees of virilization [1,4,5]. Timely removal of the gonads will prevent the virilization and risk of gonadal malignancy [75]. These clinical characteristics are similar to those of other conditions, such as partial androgen insensitivity syndrome and 5-α-reductase type 2 deficiency; therefore, misdiagnosis is not unusual for this disease [9,27,51,53,62,64,68,70]. In 17-β-HSD3 deficiency, the conversion of androstenedione to testosterone is reduced, leading to a decreased testosterone/androstenedione ratio [76]. Unfortunately, androstenedione measurements were not included in the investigation protocol at the time of the evaluation of our patient, and this led to a delay in the exact diagnosis. Furthermore, the increase in testosterone in response to hCG stimulation led to an erroneous diagnosis of partial androgen insensitivity. It was only after negative testing for *AR* and *SRD5A2* mutations that other genetic causes were searched by a WES analysis. This allowed the identification of a compound heterozygous mutation in *HSD17B3*, which led to a final diagnosis of 17-β-HSD3 deficiency.

The p.Ala203Val and p.Glu215Asp mutations identified in our patient have already been reported as pathogenic in the literature [3,13], although this is the first time that both have been found in the same patient. Previous functional studies by site-directed mutagenesis and transfection into cultured mammalian cells demonstrated that these mutations completely abolish the enzyme’s ability to convert androstenedione to testosterone [3,13]. This contrasts with other mutations, such as the common p.Arg80Gln mutation, that were shown to maintain some residual activity [3,13], and this may explain the phenotypic variability observed among patients.

17-β-HSD3 deficiency is a rare disorder and only 187 families (239 patients) have been reported along with their causative mutations (Appendix A). A total of 70 different *HSD17B3* mutations have been reported so far in the literature. However, many of these were shared by several families, suggesting the existence of either mutational hotspots or founder effects. In particular, the common p.Arg80Gln mutation has been found almost exclusively in patients originating from the Mediterranean and Middle East regions, and the common c.277+4A>T mutation has been found predominantly in patients originating from western Europe (Appendix A). Haplotype analyses of the chromosomal region of the *HSD17B3* gene have suggested that these mutations are ancient and originate from genetic founders [70]. Reported mutations were most frequently missense (55%), followed by splice-site (29%), small deletions and insertions (7%), nonsense (5%), and gross rearrangements (2%). These are all expected to lead to loss of function of the 17-β-HSD3 enzyme. Interestingly, two families were reported to have partial gene duplications [51,56], although it remains to be elucidated how these affect the enzyme activity.

In conclusion, we present a case of DSD due to 17-β-HSD3 deficiency in whom the genetic diagnosis was established through a WES analysis. We also present a comprehensive list of all published *HSD17B3* mutations that may be of use to those involved in the clinical management and genetic diagnosis of this disorder.

## 4. Materials and Methods

The genetic studies were approved by the Institutional Ethics Committee of the Faculty of Health Sciences, University of Beira Interior, Covilha, Portugal (Ref: CE-FCS-2013-017). Written informed consent was obtained from the patient’s legal guardian. Genomic deoxyribonucleic acid (DNA) was extracted from the peripheral blood leucocytes of the patient and used for a WES analysis. Targeted enrichment was performed using the Agilent V6 Exon Kit (Agilent Technologies, Santa Clara, CA, USA), and the target regions were sequenced on a DNBSEQ sequencing platform (BGI Tech Solutions, Hong Kong) with paired-end reads of 100 bp and 100 × raw read coverage. The reads were mapped to the human reference genome GRCh37/hg19 using the Burrows–Wheeler Aligner (BWA-MEM, v0.7.17) software [77,78]. Variant calling and annotation was performed using the Genome Analysis Toolkit v3.5 (GATK, v4.1.4.1) [79,80]. Variants were selected if they were cumulatively: (a) located in genes previously associated with DSDs [11]; (b) located in coding exons or adjacent splice sites; (c) non-synonymous; and (d) absent or rare (population frequency <0.1%) in the Genome Aggregation Database (gnomAD) [81]. Pathogenic variants were confirmed by conventional Sanger sequencing using a CEQ DTCS sequencing kit (Beckman Coulter, Fullerton, CA, USA) and an automated capillary DNA sequencer (GenomeLab TM GeXP, Genetic Analysis System, Beckman Coulter, Fullerton, CA, USA).

Published *HSD17B3* germline mutations were identified by searching the PubMed database (National Center for Biotechnology Information, U.S. National Library of Medicine, National Institutes of Health) (https://www.ncbi.nlm.nih.gov/pubmed, accessed on 9 June 2022) for articles, using the keywords “mutation” combined with “HSD17B3” or “17 beta hydroxysteroid dehydrogenase 3”. Reference lists of articles were also searched to identify further articles. The articles were analyzed for evidence of data duplication and patients that had been included in previous mutation studies were excluded from the analysis. Each published mutation was checked for accuracy by comparison to the *HSD17B3* wild-type sequence. Errors due to the incorrect assignment of nucleotide or codon numbers, or translation errors between codon and amino acid residues were corrected whenever possible. The mutations shown only at the amino acid level were converted to single-nucleotide changes when it was possible to predict the altered base using the genetic code. When more than one nucleotide change could account for the amino acid change or when other ambiguous changes were indicated, the precise mutation was considered unavailable. The numbering of each nucleotide was changed, whenever necessary, to comply with current recommendations for mutation nomenclature [41], whereby nucleotide +1 was the A of the ATG-translation initiation codon. The mutations were described in relation to the *HSD17B3* cDNA reference sequence (GenBank accession number NM_000197.2).

## Figures and Tables

**Figure 1 ijms-23-10026-f001:**
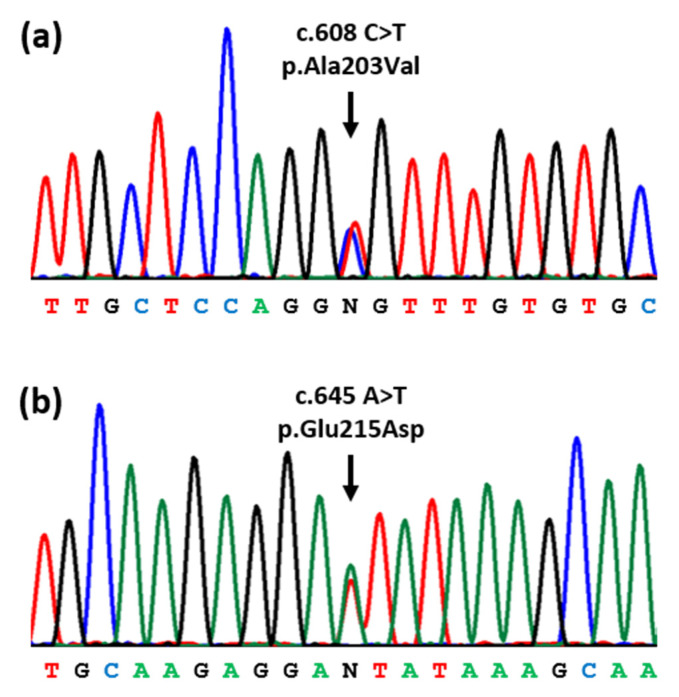
Partial DNA sequence of the 17-beta-hydroxysteroid dehydrogenase 3 (*HSD17B3*) gene. The patient was found to be compound heterozygous for two missense mutations (arrows) in exon 9: (**a**) NM_000197.2:c.608C>T, p.Ala203Val; (**b**) NM_000197.2:c.645A>T, p.Glu215Asp.

**Figure 2 ijms-23-10026-f002:**
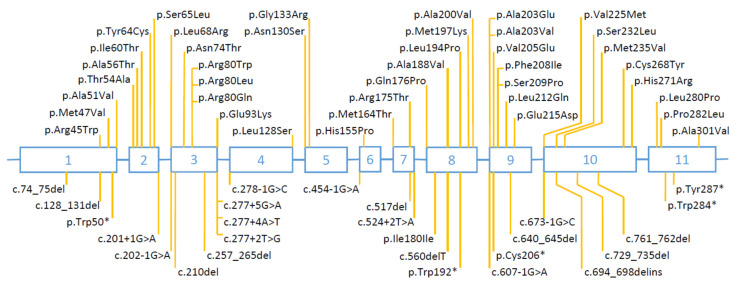
Location of mutations reported in the literature. Open boxes (numbered 1 to 11) represent coding exons of the 17-beta-hydroxysteroid dehydrogenase 3 (*HSD17B3*) gene and intervening horizontal lines represent introns (not drawn to scale). Missense mutations are depicted above the gene structure and all other mutations are represented below. Gross rearrangements (whole exon deletions and duplications) are not represented. Please refer to Appendix A for further details.

**Table 1 ijms-23-10026-t001:** Baseline and hCG-stimulated plasma hormone levels.

	Day 1 (Baseline)	Day 2	Day 4
Total testosterone	<1 ng/dL (NR: 1–20)	11.3 ng/dL	36.9 ng/dL
DHT	2.8 ng/dL (NR: 1.5–5.4)	3.2 ng/dL	6.0 ng/dL
T/DHT ratio	0.4 (NR: <10)	3.5	6.2
SHBG	85.8 nmol/L (NR: 48–142)	88.6 nmol/L	91.8 nmol/L
3α-Androstanediol	0.6 ng/mL (NR: 0.2–3.8)	0.5 ng/mL	1.0 ng/mL
Estradiol	<5 pg/mL (NR: <22)	<5 pg/mL	<5 pg/mL
Estrone	3.9 pg/mL (NR: <25)	3.1 pg/mL	5.8 pg/mL
FSH	1.7 IU/L (NR: 0.25–1.92)	-	-
LH	0.1 IU/L (NR: 0.02–1.03)	-	-

The stimulation test consisted of subcutaneous administration of human chorionic gonadotropin (hCG) 2000 IU/day, during 3 consecutive days. NR, normal range values for males of same age (based on the laboratory-specific reference ranges for the pediatric population at the time of hormone measurements); T, total testosterone; DHT, dihydrotestosterone; SHBG, sex hormone-binding globulin; FSH, follicle-stimulating hormone; LH, luteinizing hormone.

## Data Availability

The data that support the findings of this study are available from the corresponding author on reasonable request.

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
