# Peer review of "Disorder of Sex Development Due to 17-Beta-Hydroxysteroid Dehydrogenase Type 3 Deficiency: A Case Report and Review of 70 Different HSD17B3 Mutations Reported in 239 Patients"

_ijms, 2022, doi:10.3390/ijms231710026_

Round 1

Reviewer 1 Report

Authors reported a child with 17-beta-hydroxysteroid dehydrogenase type 3 (17-β-HSD3) deficiency  due to a compound heterozygous mutation of the HSD17B3 gene (c.608C>T, p.Ala203Val, 17 and c.645A>T, p.Glu215Asp). They also reviewed all HSD17B3 mutations published so far in 18 cases of 17-β-HSD3 deficiency. However, article has serious flaws and additional experiments are needed.

Reviewer 2 Report

These authors reported the case with a compound heterozygous mutation of the HSD17B3 gene and reviewed HSD17B3 mutations published so far in cases of 17-β-HSD3 deficiency. A landscape view of 70 different HSD17B3 mutations from 239 patients will be useful to diagnosis of 17-β-HSD3 deficiency. 

One comment is about the pedigree study in this family. Is there any family history related DSD? 

Round 2

Reviewer 1 Report

The study are needed to be improved by doing more experiments and functional studies. For example, they could perform RNA sequencing on patient, heterozygous carriers, and normal samples and provide the differential expression or they could provide this mutation in human cells using crispr/cas9 system and then perform functional analysis to find impaired pathways and biological processes.

Author Response

>>>Authors’ response: We thank the reviewer for his/her comments. We agree that functional experiments are important to confirm the pathogenicity of new genetic variants. However, alternative methods of investigating variants (e.g. bioinformatic analyses) are often used and there are many recent examples of mutations published in IJMS without functional experiments (e.g. IJMS 2022; 23(7):3670., IJMS 2022; 23(6):3414., IJMS 2022; 23(6):2967., IJMS 2021; 22(20):11007, IJMS 2021;22(12):6215, IJMS 2021; 22(4):2089, IJMS 2021; 22(3):1084). But the most important note is that the 2 mutations identified in our patient have already been functionally characterized in the literature (please see discussion section), so there would be no benefit in repeating these studies. We hope the reviewer will be satisfied with this explanation and with the improvements that were suggested by the other reviewers and academic editor, and that the article will be useful to the readers of this special issue on "Molecular Genetics of Disorders of Sex Development".

Round 3

Reviewer 1 Report

One point is that this study only covers report of previously known mutations and not special findings. Authors mentioned that their identified compound heterozygous mutations have been reported. Therefore, there is no very interesting finding for readers and it may be suitable for journal like Genes in MDPI.

Author Response

We understand the reviewer's point about the lack of novelty of the identified mutations. Regardless of the genetic findings, the manuscript exemplifies the "real world" clinical and diagnostic challenges that often arise in the management of these rare patients with DSD. In addition, we believe that our systematic review of all published mutations will be a useful resource to those involved in this field. We hope that readers will enjoy reading this article in the Special Issue “Molecular Genetics of Disorders of Sex Development”.